# Indirect Human Influences in Fear Landscapes: Varying Effects of Moonlight on Small Mammal Activity along Man-Made Gradients of Vegetation Structure

**DOI:** 10.3390/life13030681

**Published:** 2023-03-02

**Authors:** Alba Pasquet, Ignasi Torre, Mario Díaz

**Affiliations:** 1Department of Evolutionary Biology, Ecology and Environmental Sciences, Faculty of Biology, University of Barcelona, Av. Diagonal 643, E-08028 Barcelona, Spain; 2BiBio Research Group and Small Mammal Research Area, Natural Sciences Museum of Granollers, C/Francesc Macià 51, E-08402 Granollers, Spain; 3Department of Biogeography and Global Change (BGC-MNCN-CSIC), National Museum of Natural Sciences, C/Serrano 115 Bis, E-28006 Madrid, Spain

**Keywords:** small mammals, predation, moonlight, vegetation structure, Mediterranean habitats

## Abstract

Risk of predation is one of the main constraints of small mammal distribution and foraging activity. Aside from numerical effects on population size due to the presence and abundance of predators, indirect cues, such as vegetation structure and moonlight, determine patterns of activity and microhabitat use by small mammals. Indirect cues are expected to interact, as shading provided by vegetation can suppress the effects of changing moonlight. We analyzed the effects of moonlight levels on the activity patterns of three common small mammal species in Mediterranean habitats, and tested whether moonlight effects were modulated by shadowing associated with the development of tall vegetation due to spontaneous afforestation following land abandonment. *A. sylvaticus*, a strictly nocturnal species, decreased activity under moonlight with no interactive effects of vegetation cover. *C. russula* showed no activity change with moonlight levels and *M. spretus* increased activity, although activity in both species was mostly determined by vegetation cover, that favored it. The effects of moonlight on small mammal activity were not homogeneous among species, nor were the interactive effects of man-made gradients of habitat structure, a fact that will produce community changes along vegetation gradients mediated by varying fear landscapes.

## 1. Introduction

Surviving in nature is challenging. Animals’ exposure to predators plays a key role in species coexistence, thus structuring ecological communities [1]. Predation influences the ecology and evolution of mammals [2] and modifies prey behavior in the wild [3]. Predator effects can be direct, by physically removing individual prey, or indirect, by affecting prey behavior [4,5]. Because predation represents an important cause of death for many small mammals, it has led to a variety of predator detection mechanisms and antipredator responses by prey species. Indeed, small mammals are prey for several avian, reptilian, and mammalian predators [6], and there is ample evidence indicating that rodents alter their foraging behavior in risky situations. Rodents preferentially forage in sheltered microhabitats (thick vegetative cover), where the risk of avian predation is lower [7]. Most of the literature and theoretical framework on predation risk and small mammals has been developed in unusually risky environments, where predation risk is likely highest (e.g., deserts, [8]), but these models have been generalized and applied worldwide, from the tropics to the poles [9,10]. In the Mediterranean basin, predators’ presence (or direct cues of it) alters small mammal behavior in response to the auditory, visual, and chemosensory cues of predators [11,12]. Behavioral effects of predation include altered movement and activity patterns (shifting activity to less risky microhabitats or to less risky time periods [4]), grouping behavior, and ultimately changes in fecundity and stress levels, causing decreasing population growth rates and altering community interactions [2]. Small mammals assess a foraging cost of predation to compensate for the risk of predation, trading-off foraging and safety [6].

The direct effects of predators’ presence on prey behavior can be modulated by environmental factors which influence the perception of predation risk. These so-called indirect cues of predation risk (as they are independent of the physical presence of predators nearby) may even have stronger effects on prey behavior that direct cues [6,13]. The antipredatory cover provided by key vegetation traits and moonlight levels are the main such indirect cues [6,13], so small mammals are usually more active under vegetation cover and in the dark than in full moon nights [2]. Both cues can also interact, so that dependence on vegetation cover can be modulated by moonlight levels [13,14], or moonlight effects can be suppressed in habitats providing enough antipredatory cover [11,14].

The absence or presence of vegetation cover influences the behavior and ecology of nocturnal animals [15]. To reduce the risk of predation, small mammals can follow different strategies: foraging under vegetation cover [16], where they can find safe habitats or microhabitats [15] (Figure 1), or foraging in areas with increased lines of sight to detect predators earlier, thus increasing their chances of escape [16]. Foraging rodents attempt to balance foraging activity and safety by spending less time under high levels of predation risk [15]. On the other hand, moonlight has also been shown to be an influential factor, making organisms either more conspicuous or detectable to their predators [14,17]. In the case of small mammals, especially rodents [15], the increased risk of predation under high moonlight levels causes them to avoid illuminated nights and open areas, favoring reducing foraging activity under the full moon and a preference for sheltered microhabitats [13,17], although higher levels of activity have also been observed for some rodent species during bright nights [2,18].

The interactive effects of vegetation structure and moonlight on small mammal activity implies a potential role of human land uses on indirect cues shaping fear landscapes (Figure 1). In the Mediterranean basin, changes in socio-economic conditions throughout the last decades produced the cessation of traditional land use practices. These changes are causing the conversion of open-land uses (crops, grasslands, scrub) to forest habitats, following a natural rewilding process (e.g., a passive restoration of natural ecosystems through ecological succession [19]). Gradients of vegetation structure complexity are interesting grounds for studying small mammal population responses, due to the strong habitat preferences shown by both common small mammal species and their predators [20,21]. Vegetation gradients are expected to produce significant changes in abundance and activity that can be modulated by perceived predation risk. These, in turn can be modulated under different levels of moonlight exposure. In forests, high canopy cover and low covers of short shrubs imply high risks for small mammals. This high exposure leads to a high perceived predation risk, together with high predation pressure (more forest predators that prey on small mammals [22]). However, lunar illumination can be mostly intercepted by the tree layer, therefore reducing the risk of predation even during full moon periods [11]. Scrublands would be better habitats for small mammals, as the high cover of short shrubs and lower abundances of small mammal predators reduce predation risk. However, lunar illumination reaches the lower vegetation levels where small mammals live, so that the perceived predation risk can be modulated by the moon cycle. Strong responses of abundance/activity to changing moonlight are thus expected in scrubland but not under dense forest canopies.

We analysed how vegetation structure gradients due to land use changes modulate the responses of small mammals to changing moonlight. We expect that: (1) both moonlight and vegetation cover will affect the presence/activity of common small mammals; (2) these effects will be modulated by vegetation structure, with stronger responses in open habitats than in dense forests; and (3) both pure and interactive effects will differ among small mammal species with different dependency of antipredatory cover and/or sensitivity to nocturnal predators. If this happens, man-made changes in fear landscapes mediated by gradients of vegetation structure may drive small mammal responses to changing land uses in the Mediterranean region.

## 2. Materials and Methods

### 2.1. Study Area

Field work was conducted in the natural parks of Collserola, Garraf, Serralada Litoral, Serralada de Marina, Montnegre and Sant Llorenç del Munt i l’Obac (Catalonia, Spain) (Figure 2). The sampled habitats were Holm oak (*Quercus ilex* L.) woodlands, scrublands (garrigue, maquis and scrub), and pine forests (*Pinus halepensis* Mill. and *Pinus pinea* L.). Small mammal monitoring was conducted in seventeen study plots between 2008 and 2018. The plots were established in the most representative habitats of each area [22].

### 2.2. Small Mammal Sampling

The sampling design followed the SEMICE long-term sampling protocol [22] https://www.semice.org/, accessed on 15 January 2022). Sampling plots were 75 m × 75 m squares defined by a 6 × 6 trap capture grid composed of 18 Sherman live traps and 18 Longworth live traps located in alternated positions [23] and separated by 15 m. Traps were located under the cover of shrubs, rocks, leaflets or herb tussocks. They were baited with a mixture of oil-and-tuna and a slice of apple and were provided with a damp-proof cotton ball to increase thermal insulation. Each sampling session consisted of three consecutive nights of trap exposure, with trap checks during the early morning. Sessions were repeated twice a year (every six months, in spring and autumn). Captured individuals were weighed, sexed, marked with permanent ear tags in the case of rodents and with fur marks in the case of shrews, and were released at the same catching site [22].

Vegetation structure, rather than vegetation composition, is key for small mammals [24]. We estimated the vegetation structure of sampling plots by *ALS LiDAR* [20] obtained from the Institut Cartogràfic i Geològic de Catalunya (flights 2016–2017). *LiDAR* provides maps of the three-dimensional structure of vegetation, even for complex and heterogeneous habitats (forests with several vegetation strata) [20,21], showing similar or better performance than visual field-based techniques [25,26] The extraction of the *LiDAR* data was performed on circular plots with a radius of 53 m (8834 m^2^) centred on each plot, being equivalent to the square of 75 m × 75 m delimited by the four corners of the SEMICE sampling plots. LiDAR point clouds were obtained with a point density between 1 and 4.28 points/m^2^. Since common small mammals are ground-dwellers or only showed moderate arborealism [20,27], we calculated the relative contribution of the vegetation layers below 1 m (RCV < 1 m), which provides shelter to small mammals. As a measure of overall forest structure, we calculated the relative contribution of the topmost layer (RCV > 2.5 m), which represents the cover of tree and shrub canopies.

Moonlight levels can be obtained with different quality and resolution: from fine-grained direct measures of moonlight reaching the ground obtained in situ by means of luxometers [28,29], to coarse-grained indirect categorical levels obtained from almanacs [30]. We used the last method, and moonlight levels were extracted for every plot and sampling session by retrospectively gathering moon phases from Internet databases (https://www.tutiempo.net/luna/fases.htm, accessed on 15 February 2022). The levels were transformed into a categorical variable with values ranging from zero (new moon) to 10 (full moon), an approach also used in other small mammal studies [31,32]. Moonlight levels arriving at the 1 m-tall vegetation level were computed by subtracting from 100 (total moonlight level) the relative contribution of the layers above 1 m (relative cover of vegetation: RCV > 1 m).

### 2.3. Data Analysis

Generalized Linear Mixed Models [33] were used to estimate the effects of moonlight levels on small mammal abundance, estimated as the total number of captures (including recaptures). Although abundance is a population parameter determined by longer-term processes than changing lunar illumination, we used the total number of captures as a proxy for activity–detectability, since numbers with and without recaptures are strongly correlated for the common small mammal species analysed here [22]. Furthermore, abundance estimates based on total catches of individuals—including recaptures—will be more representative of small mammal activity.

Three models have been tested, one for each species, in which the abundance was considered as the response variable. We used four predictors: moonlight levels (interval scale from zero to 10); the relative cover of vegetation above 2.5 m (RCV > 2.5 m), as an indirect measure of moonlight blocked by vegetation canopies; the relative cover of vegetation below 1 m (RCV < 1 m), as a measure of refuge against small mammal predators; and the sampling period (spring and autumn). Due to manpower limitations, sampling sessions were carried out in weeks differing in the phase of the moon cycle. Differences among plots other than those due to moonlight levels were controlled for by including plot as a random factor, which was nested within the habitat (shrubland or forest) to account for uncontrolled factors related to habitats that were not associated to vegetation cover. Year was also included as a random factor to account for interannual variations in abundance. Models were classified according to their AICc and tested using Likelihood ratio tests (LRTs). Submodels were tested considering all the fixed (the four predictors) and random factors (plot and year) altogether (submodel 1), calculating the variance explained by all the fixed factors (marginal R^2^) and for the whole model (conditional R^2^). We then tested one submodel for each predictor or fixed factor (models nº 2, 3, 4 and 5), to calculate the variance explained by each independent factor. The contribution of each fixed factor was calculated by subtracting the marginal variance (marginal R^2^) with the conditional variance (conditional R^2^) (variance explained by the whole model consisting of one fixed and two random factors).

Analyses were performed under *rStudio* [34] using the *MuMin* and *lme4* packages [35]. The functions used were *lmer* and *glmer* (used to fit a linear mixed model) and *dredge* (generate a model selection table of models with combinations of fixed effect terms in the global model). We calculated pseudo-R^2^ values [36] by means of the function *r.squaredGLMM* and the delta method for variance estimation.

## 3. Results

We caught 1502 wood mice *Apodemus sylvaticus* L., 754 greater white-toothed shrews *Crocidura russula* Hermann, and 361 Algerian mice (*Mus spretus* Lataste). Other common small mammals in Mediterranean landscapes, such as the garden dormouse (*Eliomys quercinus* L.) and the bank vole (*Clethrionomys glareolus* Schreber), were too scarce during sampling for analyses (just three dormouse and nine voles); therefore, we only tested responses to vegetation and moonlight of the three most abundant species listed above.

Differences in vegetation structure of plots obviously affected the proportion of moonlight reaching the vegetation 1 m above ground. Such proportions were larger than 60% (81.6% on average, up to 100% in some plots) in scrubland plots, where most vegetation cover and biomass accumulated at heights below 1 m. Canopy covers in forest plots block moonlight, so that proportions reaching 1 m above ground were lower than 30%. Higher canopy development in holm oak woodlands than in pinewoods produced higher shading (4.9% vs. 21.3% moonlight reaching the 1-m level for oaks woodlands and pinewoods, respectively).

The number of captures of each of the three small mammal species were almost completely explained by the fixed and random factors included in the GLMM models (saturated models: 91–99%, Table 1). Vegetation structure was the most important factor for *C. russula* (55%) and *M. spretus* (54%), whereas season was more relevant for *A. sylvaticus* (57%). The effect of varying moonlighting on the abundance of each species was low but significant overall (0.1–2.3%) (Table 2, Figure 3).

The best model for *A. sylvaticus* showed an overall negative influence of moonlight on captures (Table 2, Figure 4), no influence of vegetation cover, and strong effects of seasonality (more captures in spring than in autumn). No interaction between vegetation structure and moonlight was detected, and moonlight effects were stronger in autumn than in spring (significant moonlight * period interaction; Table 2). The best model for *C. russula* showed a negative influence of tall vegetation cover on captures, no moonlight effects (Figure 4), and more captures in autumn that in spring. Furthermore, no interaction between vegetation structure and moonlight was detected. The best model for *M. spretus* also showed a negative influence of tall vegetation, positive effects of moonlight on captures, and a negative effect of tall vegetation cover on moonlight effects (weaker effects under cover) (Figure 4). Captures increased from spring to autumn, as in shrews, and moonlight effects were stronger in spring than in autumn, contrary to the results of the *Apodemus* model.

## 4. Discussion

Common small mammals showed heterogeneous responses to perceived predation risk along man-made gradients of the vegetation structure. Half of the variance of small mammal abundance (55–57%) was explained by the random factors of site and year, as somewhat expected from strong interannual variability in small mammal population sizes and plot effects not considered here [37,38]. Rather surprisingly, indirect cues of perceived predation risk (moonlight) were almost irrelevant (<1% of variance explained), although part of this low sensitivity may be due to the use of abundance as a proxy for activity. Response variables more directly linked to behavioral responses to risks (e.g., giving-up densities, the amount of food eaten [8,16]) may be more sensitive. On the other hand, the use of indirect measures of actual moon light arriving to the ground level (e.g., LiDAR variables combined with moon phase levels taken from almanacs) could also have contributed to low sensitivity. Direct estimates of the moonlight reaching the strata where small mammals forage using luxometers may increase sensitivity [28]. Finally, we did not consider cloud cover during the study nights because this data was not available; reduced moonlight levels on cloudy nights will also have contributed to lowering the sensitivity of this study.

### 4.1. Moonlight Affects the Activity of Common Small Mammals

Responses by small mammals to varying moonlight were species-specific rather than general [16], and were weaker than expected. The results for wood mice fitted to the expected negative relationship between moonlight levels and rodent activity, whereas white-toothed shrew showed no responses and Algerian mice showed an unexpected increase in activity with increasing moonlight. The wood mouse is a primarily nocturnal species that starts its activity at sunset and retreats to refuges at dawn, thus avoiding periods of higher illumination as well as the brighter phases of the moon cycle [39]. This is also the case for *Apodemus* spp. in general and for other strictly nocturnal species, [32,40] Wood mice change their daily foraging schedule in response to the activity of predators such as genets [11], but this behavior was not affected by moonlight. The effects of moonlight in this study were significant, although low variance explained the indicated low relevance. Greater white-toothed shrews and Algerian mice, although nocturnal, usually extend the foraging activity to daytime. Shrews are active all year round, both at night and during the day in the areas where they do not hibernate [41,42], while Algerian mice displayed diurnal activity at least during winter [43]. The effects of moonlight on small mammal activity are thus expected to vary among species according to their degree of diurnal activity. The strictly nocturnal wood mice showed the expected negative effect of moonlight on activity, whereas more diurnal species would be more tolerant to moonlight [40]. Their activity did not decrease under full moon nights, or even they might profit from foraging in areas with increased sightlines to spot predators earlier, thereby increasing their chances to escape [16].

### 4.2. Moonlight Effects Modulated by Vegetation Structure. Comparison of Open/Closed Habitats

The vegetation structure of habitats obviously influences the quantity of lunar illumination that reaches the vegetation layers that small mammals inhabit. Fine-scale changes in the vegetation structure at this level alter the perception of fear by small mammals, and these responses seem to depend on the abundance of predators [13,21]. The perception of risk seems to be higher in open areas such as scrublands [16], where the proportion of lunar illumination reaching the lower vegetation levels is much greater than in forests.

The responses of small mammals to the relative cover of vegetation (RCV) and its interaction with moonlight were also species-specific. Wood mice abundance did not vary along the vegetation gradients from open (scrubland) to closed (forest) plots, as somewhat expected from their generalist habitat use [20]. The lack of interactive effects of moonlight and habitat structure indicated that perceived predation risk was independent from vegetation structure, and was not modulated by indirect cues of predation risk [11]. Greater white-toothed shrews and Algerian mice showed a negative response to the RCV of tree canopies. Both species showed significant associations with open-scrubland habitats in the area [20], thus suggesting a role of predator abundance, which was lower in scrublands, as well as of the protective cover of short vegetation. Shrews have been shown to reduce foraging activity under the presence of terrestrial predators [44] (direct cues), but our results suggested a lack of response to indirect cues of predation risk (vegetation cover and moonlight). The preference for plots covered with short vegetation altogether with behavioral traits (using torpor and diurnal activity) will reduce encounters with predators and, hence, decrease perceived predation risk [45]. Rather surprisingly, the Algerian mouse showed a positive effect of moonlight on its abundance/activity. This species reduced or shifted its foraging activity under predation risk [12,46], but in our case, the indirect cues of perceived predation risk increased abundance/activity. The cover of tall vegetation decreased the effects of moonlight for this species: Algerian mice respond less to indirect cues of risk in open habitats, where small mammal predators are more scarce [11,47].

### 4.3. Conclusions

The responses of three common small mammal species to indirect cues of perceived predation risk were heterogeneous. Three different responses were detected: (1) a negative and consistent effect of an indirect cue of risk (moonlight) that was unaffected by another indirect cue (vegetation profiles), in the wood mouse; (2) a positive response to moonlight in the Algerian mouse that was modulated by vegetation profiles, with stronger effects in the areas with less cover of tall canopies where predators are actually absent (open habitats); and (3) a lack of responses in the Greater white-toothed shrew that could be associated to microhabitat selection (preferred plots with more cover of short vegetation) and behavioral adaptation (diurnal activity) to avoid predation. The effects of moonlight on small mammal activity were not homogeneous among species, nor were the interactive effects of gradients of habitat structure, a fact that will produce community changes along vegetation gradients mediated by varying fear landscapes. Therefore, man-made changes in fear landscapes mediated by gradients of vegetation structure may drive small mammal responses to changing land uses in the Mediterranean region.

## Figures and Tables

**Figure 1 life-13-00681-f001:**
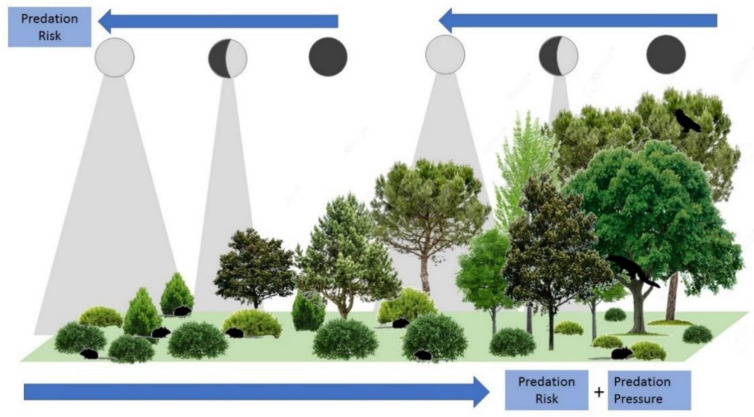
Idealized model of predation risk and predation pressure for small mammals along vegetation structural gradients in Mediterranean landscapes, showing the potential interactive effects between vegetation cover and lunar illumination.

**Figure 2 life-13-00681-f002:**
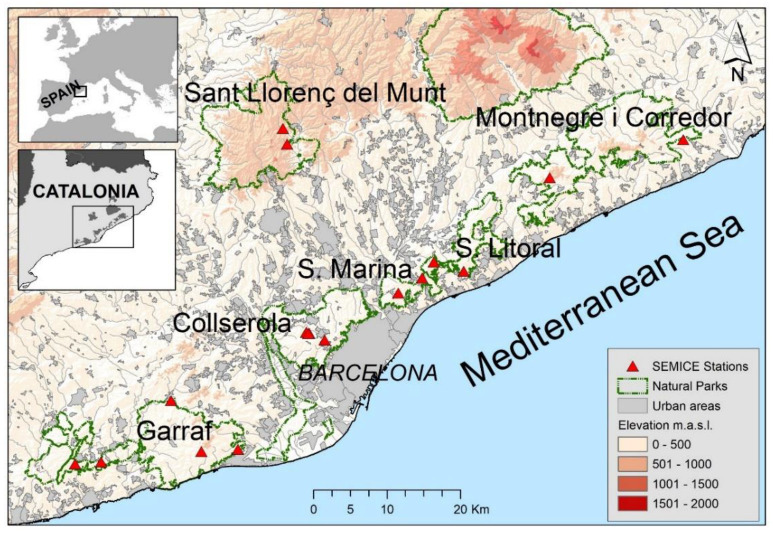
Map with the locations of 17 SEMICE sampling stations within the six natural parks in Barcelona province (Catalonia, Spain) where the field work was conducted.

**Figure 3 life-13-00681-f003:**
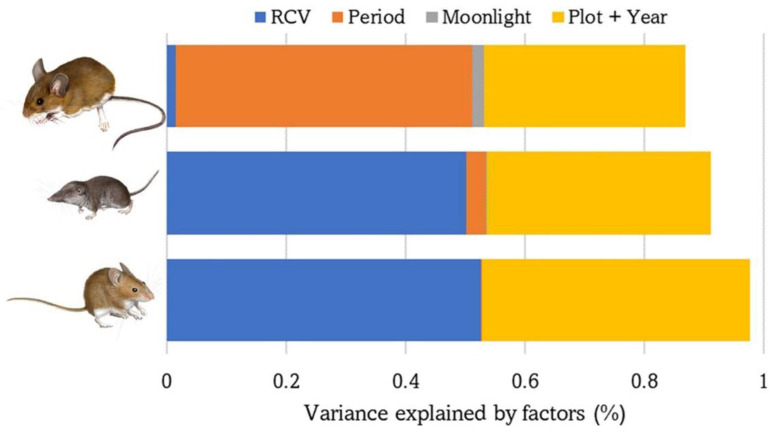
Variance explained by the GLMM models performed with the abundance/activity of the three species of small mammals based on the following explanatory variables: vegetation cover (RCV), sampling period (spring or autumn), moonlight, plot and year. From top to bottom: *A. sylvaticus*, *C. russula*, and *M. spretus*.

**Figure 4 life-13-00681-f004:**
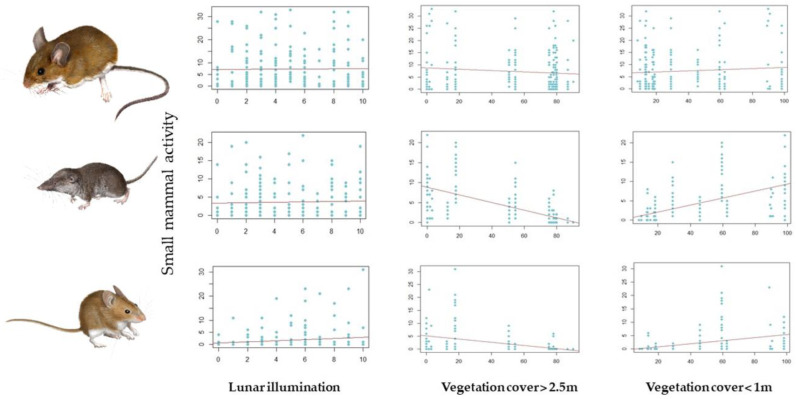
Changes in species abundance/activity depending on the fixed factors considered in the GLMM models: lunar illumination (classified at 10 levels of illumination, where 0 is complete darkness and 10 is a full moon) and the relative contribution of vegetation cover above 2.5 m and below 1 m (RCV). From top to bottom: *A. sylvaticus*, *C. russula*, and *M. spretus*.

**Table 1 life-13-00681-t001:** GLMM models testing for the influence of moonlight, relative vegetation cover below 1 m and above 2.5 m height, and season, on the abundance of three common small mammal species. The marginal R^2^ explains the variability of fixed factors (illumination, RCV, and period) and the conditional R^2^ explains fixed and random factors (the plot by habitat and the year).

Models		*Apodemus sylvaticus*	*Crocidura russula*	*Mus spretus*
Mod. nº 1. Abundance ~ Moonlight + Period + RCV > 2.5 m + RCV < 1 m	Marginal R^2^	0.50	0.54	0.55
	Conditional R^2^	0.91	0.95	0.99
Mod. nº 2. Abundance ~ Moonlight	Marginal R^2^	0.02	0.00	0.00
	Conditional R^2^	0.84	0.95	0.98
Mod. nº 3. Abundance ~ Period	Marginal R^2^	0.50	0.03	0.00
	Conditional R^2^	0.91	0.95	0.98
Mod. nº 4. Abundance ~ RCV > 2.5 m	Marginal R^2^	0.02	0.50	0.52
	Conditional R^2^	0.82	0.95	0.98
Mod. nº 5. Abundance ~ RCV < 1 m	Marginal R^2^	0.01	0.44	0.47
	Conditional R^2^	0.82	0.95	0.98
Mod. nº 6. Abundance ~ Moonlight * RCV > 2.5 m	Marginal R^2^	0.05	0.50	0.53
	Conditional R^2^	0.84	0.95	0.98
Mod. nº 7. Abundance ~ Moonlight * RCV < 1 m	Marginal R^2^	0.05	0.44	0.48
	Conditional R^2^	0.84	0.95	0.98
Mod. nº 8. Abundance ~ Moonlight * Period	Marginal R^2^	0.48	0.03	0.01
	Conditional R^2^	0.92	0.95	0.98

**Table 2 life-13-00681-t002:** Best GLMM models fitted to numbers of captured individuals of each species, based on the four predictor variables and their interaction with moonlight.

	*Apodemus sylvaticus*	*Crocidura russula*	*Mus spretus*
Moonlight	−0.06 *** (0.01)	-	0.20 *** (0.04)
RCV > 2.5 m	-	−0.04 *** (0.01)	−0.04 ** (0.01)
RCV < 1 m	-	-	-
Period (Autumn)	−1.66 *** (0.11)	0.60 *** (0.01)	1.08 *** (0.24)
Moonlight * RCV > 2.5 m	-	-	−0.00 *** (0.00)
Moonlight * RCV < 1 m	-	-	-
Moonlight * Period	0.04 * (0.02)	-	−0.13 ** (0.04)

***: *p* < 0.001; **: *p* < 0.01; *: *p* < 0.05.

## Data Availability

The data presented in this study are available on request from the corresponding author.

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
