# Peer review of "Indirect Human Influences in Fear Landscapes: Varying Effects of Moonlight on Small Mammal Activity along Man-Made Gradients of Vegetation Structure"

_life, 2023, doi:10.3390/life13030681_

Round 1

Reviewer 1 Report

Reading through the manuscript, I realised a potential weakness in the study concerning the measurement of moonlight at ground level. The authors state that they did not directly measure the light reaching the ground but referred to an almanac. Although this is a widely used practice, especially in the past in studies that retrospectively analyse the effects of moonlight on the biology of organisms, I believe that especially under the experimental conditions of the study, this approach can be misleading. The small mammals under study are ground-dwellers and find shelter under vegetation <1m high. From direct experience (actual measurement of moonlight with a luxmeter), no moonlight passes through the forest canopy (Vignoli et al., 2014), let alone if there are different layers of vegetation covering the ground. Furthermore, the source used by the authors to estimate moonlight intensity does not seem to take into account cloud cover, which is one of the main factors determining the amount of night light reaching the ground. I therefore wonder how one can correctly approximate moonlight intensity without taking all these factors into account. I recommend the authors to read the discussions on this topic in Vignoli and Luiselli (2013), in Onorati and Vignoli (2017), and in Bissattini et al. (2020) who analyse the problems that arise from not directly measuring the moonlight reaching the ground. I believe that such potential weaknesses of the study should at least be acknowledged by the authors in their methods and discussion.

Cited literature:

Vignoli, L., & Luiselli, L. (2013). Better in the dark: two Mediterranean amphibians synchronize reproduction with moonlit nights. Web Ecology, 13(1), 1-11. Vignoli, L., D’Amen, M., Della Rocca, F., Bologna, M. A., & Luiselli, L. (2014). Contrasted influences of moon phases on the reproduction and movement patterns of four amphibian species inhabiting different habitats in central Italy. Amphibia-Reptilia, 35(2), 247-254. Onorati, M., & Vignoli, L. (2017). The darker the night, the brighter the stars: consequences of nocturnal brightness on amphibian reproduction. Biological Journal of the Linnean Society, 120(4), 961-976. Bissattini, A. M., Buono, V., & Vignoli, L. (2020). Moonlight rather than moon phase influences activity and habitat use in an invasive amphibian predator and its native amphibian prey. Acta oecologica, 103, 103529.

Author Response

Reading through the manuscript, I realised a potential weakness in the study concerning the measurement of moonlight at ground level. The authors state that they did not directly measure the light reaching the ground but referred to an almanac. Although this is a widely used practice, especially in the past in studies that retrospectively analyse the effects of moonlight on the biology of organisms, I believe that especially under the experimental conditions of the study, this approach can be misleading. The small mammals under study are ground-dwellers and find shelter under vegetation <1m high. From direct experience (actual measurement of moonlight with a luxmeter), no moonlight passes through the forest canopy (Vignoli et al., 2014), let alone if there are different layers of vegetation covering the ground.

Authors: first, we want to acknowledge to the reviewer for providing the results of his investigations on amphibia published in the four papers mentioned, which are very interesting and were used to set with more comprehensiveness the theoretical framework involving moon-activity relationships in animals. We are very grateful for your kind comments, and on the way of improving our paper. Our results point out a very marginal effect of moonlight on small mammal activity (<1% of variance in the models of the three species analysed), which could be related to the reduced amount of light arriving to the floor in the habitats analysed, even during full moon phases. We agree with your comment “From direct experience (actual measurement of moonlight with a luxmeter), no moonlight passes through the forest canopy (Vignoli et al., 2014), let alone if there are different layers of vegetation covering the ground.” Indeed, we are providing the first strong evidence (with a large and temporally expanded sample) of the lack of effects of moon light on small mammal activity. The effects of moon light on animal activity can be expected to be relevant in areas without vegetation cover, such as deserts, in which the theoretical framework of small mammals-moon light was developed. This can be also applied to open habitats, such as the ponds you studied, in which lunar illumination reaches the ground/water without any interference. We agree, using a luxmeter will be a better approach than using indirect vegetation profiles, and we were more cautious at providing strong credibility to our results. But we used for the first time (as far as we know) information on vegetation structure profiles provided by Lidar technology, which can inform about vegetation structure with similar performance as more traditional visual estimates. We are confident that the vegetation structure profiles of tall canopies cannot be properly assessed by visual estimates because the lower vegetation strata avoid seeing these parts of the forest.

Furthermore, the source used by the authors to estimate moonlight intensity does not seem to take into account cloud cover, which is one of the main factors determining the amount of night light reaching the ground. I therefore wonder how one can correctly approximate moonlight intensity without taking all these factors into account.

Authors: we agree, cloud cover can be relevant in reducing moonlight levels, but the information of weather was systematically recorded altogether with the small mammal samples from 2016 onwards (when the web of the project SEMICE was set), but unfortunately it was unavailable for eight out of the eleven years of samples used in this study (2008-2015). Nonetheless, we want to stress that some plots were paired (8 out of 17), so we have direct comparison of small mammal activity in structurally contrasted habitats, such as scrubland and forest, in the same dates and areas for half of the samples analysed (104 out 205). This means that, even different cloud cover can affect small mammal activity, we guess that under dense canopies this effect can be minor, and maybe affecting more the scrubland. Anyway, overall effects of moonlight were very reduced and almost non-existent irrespective of habitat structure, so we think that the role of cloud cover could be acting in the direction of lowering even more these effects.

 I recommend the authors to read the discussions on this topic in Vignoli and Luiselli (2013), in Onorati and Vignoli (2017), and in Bissattini et al. (2020) who analyse the problems that arise from not directly measuring the moonlight reaching the ground. I believe that such potential weaknesses of the study should at least be acknowledged by the authors in their methods and discussion.

Authors: thanks for the references provided, we centred our research on small mammals, so all those recommended citations were overlooked. However, we had a look to them to have a wider scope of the problem and having a more realistic vision of the potential biases and weakness of our results. We acknowledge the use of these references which were especially suitable for improving the theoretical framework and discussion of our article.  

 Cited literature:

Vignoli, L., & Luiselli, L. (2013). Better in the dark: two Mediterranean amphibians synchronize reproduction with moonlit nights. Web Ecology13(1), 1-11. Vignoli, L., D’Amen, M., Della Rocca, F., Bologna, M. A., & Luiselli, L. (2014). Contrasted influences of moon phases on the reproduction and movement patterns of four amphibian species inhabiting different habitats in central Italy. Amphibia-Reptilia35(2), 247-254. Onorati, M., & Vignoli, L. (2017). The darker the night, the brighter the stars: consequences of nocturnal brightness on amphibian reproduction. Biological Journal of the Linnean Society120(4), 961-976. Bissattini, A. M., Buono, V., & Vignoli, L. (2020). Moonlight rather than moon phase influences activity and habitat use in an invasive amphibian predator and its native amphibian prey. Acta oecologica103, 103529.

Reviewer 2 Report

The MS is a nice and useful piece of scientific literature, which deserves to be published.

However, some revisions are required:

1.       Although I am not a native English speaker, I strongly recommend authors to send their MS to a native speaker for language polishing.

2.       Literature search is lacking some works on effects of moonlight on behavioural ecology of rodents, including those conducted by my research group in Central Italy, which can be used as comparison and reinforce your results. Particularly, those on Savi’s pine vole and Apodemus/Myodes also reported timing of direct captures and habitat analyses, besides temporal ones.

3.       In the study area, the scientific names of plants should report the names of plants should also include the names of descriptors.

4.       Results are clear, and methods are replicable.

5.       Discussion is well-written, but a conclusion is lacking. Please, include in your discussion which of your predictions were fulfilled and which ones not.

Author Response

The MS is a nice and useful piece of scientific literature, which deserves to be published.

However, some revisions are required:

  1. Although I am not a native English speaker, I strongly recommend authors to send their MS to a native speaker for language polishing.

Authors: we let the article to be reviewed by an English native speaker

  1. Literature search is lacking some works on effects of moonlight on behavioural ecology of rodents, including those conducted by my research group in Central Italy, which can be used as comparison and reinforce your results. Particularly, those on Savi’s pine vole and Apodemus/Myodes also reported timing of direct captures and habitat analyses, besides temporal ones.

Authors: We are sorry for the partial search of related literature. We found some of the articles you mentioned, and we read and cited them to have a wider knowledge of the moon light-small mammal activity relationships.

Dell’Agnello, F., Martini, M., Mori, E., Mazza, G., Mazza, V., Zaccaroni, M., 2020. Winter activity rhythms of a rodent pest species in agricultural habitats. Mammal Res. 65, 69–74. doi:10.1007/s13364-019-00443-4

Mori, E., Sangiovanni, G., Corlatti, L., 2020. Gimme shelter: The effect of rocks and moonlight on occupancy and activity pattern of an endangered rodent, the garden dormouse Eliomys quercinus. Behav. Processes 170, 103999. doi:10.1016/j.beproc.2019.103999

Viviano, A., Scarfò, M., Mori, E., 2022. Temporal Partitioning between Forest-Dwelling Small Rodents in a Mediterranean Deciduous Woodland. Animals 12, 279. doi:10.3390/ani12030279

  1. In the study area, the scientific names of plants should report the names of plants should also include the names of descriptors.

Authors: We included the name of descriptors of the plants cited, and also for the small mammal species.

  1. Results are clear, and methods are replicable.

Authors: Thanks!!!

  1. Discussion is well-written, but a conclusion is lacking. Please, include in your discussion which of your predictions were fulfilled and which ones not.

Authors: The last paragraph can be considered as the conclusion, but we made it more clear in the revised version.

Reviewer 3 Report

This manuscript is of undoubted interest to a wide circle of zoologists. Despite the considered parameters being to a considerable degree relative, we can see the heterogeneity of the reaction of different species to environmental influences. They can be interpreted as indirect signs of predation risk, or they can be interpreted simply as a number of factors affecting the mobility and mortality of individuals. In general, I liked the manuscript, but there are some questions and comments.

relative cover of vegetation, RCV: judging by the context, this term refers not only to the height of vegetation, but also to the area covered by grass and crown density of the trees at the height above 2.5 m and below 1 m. Is that so? In that case, was the parameter of the said area and crown density evaluated in points, or in some other way? The RCV gradations should be described more clearly.

Judging by the descriptions, the authors did not take into account the peculiarities of the species composition of the grass and shrub tiers, i.e. the authors for the sake of convenience suppose that in all oak forests the canopy development is the same, it differs from brushwoods and pine forests. Why are the features of the species composition not taken into account? Plant species with a similar RCV effect on satellite photos may have different food value and productivity, which will invariably entail differences in the feeding activity of both granivorous and insectivorous species.

What do the asterisks ** , *** mean in the table. 2? Is it the level of statistical significance? It is necessary to explain this in a note to the table.

It is unclear whether the authors took into account the different abundance levels for the three species and the possible impact of the migration of young individuals. Judging by the number of captured individuals, the abundance of Apodemus sylvaticus was significantly higher than Crocidura russula and even more than Mus spretus. How do the authors interpret this? Is it really a higher abundance, or a higher activity of feeding behavior, or active migration of juveniles?

How do the authors explain the pronounced seasonal fluctuations in the abundance of Apodemus sylvaticus? Did this species have more active reproduction in the winter months than in summer? Or is it due to higher mobility in search of food in spring? Can the indifference to moonlight found in this species be associated with a shortage of food resources at a high population density?

Some of these questions are not directly related to the assessment of the influence of moonlight on the activity of small mammals, but due to the lack of explanations, there is a feeling of loose ends and weakness of the evidence base. Of course, we admit that the authors themselves repeatedly point out in the text of the manuscript that the parameters under consideration are, to some degree, relative.

Nevertheless, I believe that the manuscript should be published after making small additions.

Kind regards

Author Response

This manuscript is of undoubted interest to a wide circle of zoologists. Despite the considered parameters being to a considerable degree relative, we can see the heterogeneity of the reaction of different species to environmental influences. They can be interpreted as indirect signs of predation risk, or they can be interpreted simply as a number of factors affecting the mobility and mortality of individuals. In general, I liked the manuscript, but there are some questions and comments.

relative cover of vegetation, RCV: judging by the context, this term refers not only to the height of vegetation, but also to the area covered by grass and crown density of the trees at the height above 2.5 m and below 1 m. Is that so? In that case, was the parameter of the said area and crown density evaluated in points, or in some other way? The RCV gradations should be described more clearly.

Authors: The RCV means the relative contribution of the vegetation layer between two height levels, as the vegetation cover at heights above 2.5m, which we used to have indirect evidence of the moon light that could reach the ground level. This variable ranges from zero (when no trees are present) to > 90% in very dense forests. Also, the RCV < 1m is the cover that vegetation provides to small mammals at the ground level (0-1m). Both variables were measured on circular plots centred on each sampling plot, with point clouds obtained with a discrete return LiDAR sensor and a point density between 1 and 4.28 points/m2 (flights 2016–2017). There were different densities of points at the different elevation strata, and these were transformed to contributions or percentages over the total points registered by stratum and elevation heights. We have now explained this better in the MS (lines 152-153).

Judging by the descriptions, the authors did not take into account the peculiarities of the species composition of the grass and shrub tiers, i.e. the authors for the sake of convenience suppose that in all oak forests the canopy development is the same, it differs from brushwoods and pine forests. Why are the features of the species composition not taken into account? Plant species with a similar RCV effect on satellite photos may have different food value and productivity, which will invariably entail differences in the feeding activity of both granivorous and insectivorous species.

Authors: We agree, different vegetation composition will yield different productivity and effects on small mammal activity. However, we think that for the same vegetation community, productivity will change with seasons and years, so we will need to have additional measures of productivity (e.g., NDVI, etc.). As we stated below, it is not the scope of this article to study the effects of vegetation composition, structure, or productivity, on population dynamics. Furthermore, we did not take vegetation composition into account because generalist small mammals are mostly responding to vegetation structure rather than vegetation composition (see Garden, J.G., Mcalpine, C.A., Possingham, H.P., Jones, D.N., 2007. Habitat structure is more important than vegetation composition for local-level management of native terrestrial reptile and small mammal species living in urban remnants: A case study from Brisbane, Australia. AUSTRAL Ecol. 32, 669–685. doi:10.1111/j.1442-9993.2007.01750.x). Maybe, our approach can be an oversimplification of the habitat-small mammals’ relationships, but we expected that vegetation height and volume (measured by three-dimensional Lidar variables) could be a proxy for small mammals’ food and safety (all being equal, high vegetation values means more food and refuge available), as was yet analysed and confirmed in previous investigations using either direct and indirect vegetation cover estimates:

Torre, I., Diaz, M., Martinez-Padilla, J., Bonal, R., Vinuela, J., Fargallo, J.A., 2007. Cattle grazing, raptor abundance and small mammal communities in Mediterranean grasslands. Basic Appl. Ecol. 8, 565–575. doi:10.1016/j.baae.2006.09.016

Torre, I., Jaime-González, C., Díaz, M., 2022. Habitat Suitability for Small Mammals in Mediterranean Landscapes: How and Why Shrubs Matter. Sustainability 14, 1562. doi:10.3390/SU14031562

What do the asterisks ** , *** mean in the table. 2? Is it the level of statistical significance? It is necessary to explain this in a note to the table.

Authors: we included the significance codes at the end of the table.

It is unclear whether the authors took into account the different abundance levels for the three species and the possible impact of the migration of young individuals. Judging by the number of captured individuals, the abundance of Apodemus sylvaticus was significantly higher than Crocidura russula and even more than Mus spretus. How do the authors interpret this? Is it really a higher abundance, or a higher activity of feeding behavior, or active migration of juveniles?

Authors: after studying the populations of these three species over decades, we are confident about those differences in abundance/activity represented true abundances in the field. A.sylvaticus was the most abundant because it is a generalist “all terrain” small mammal present in all habitats, without particular habitat requirements. Indeed, this species showed the highest distribution breath in the project SEMICE (present in almost all the plots sampled). C. russula is also a habitat generalist species, but it feeds on invertebrates (higher trophic level than A.sylvaticus) but shows preference for open habitats with well-developed cover at ground level where it reaches the highest abundance. This species is less abundant than the wood mouse. M.spretus is a habitat specialist, and it is present almost exclusively in open habitats and lacking from forests. So, the order of the three species abundances obtained in this study agree with their expected habitat requirements and feeding preferences.

How do the authors explain the pronounced seasonal fluctuations in the abundance of Apodemus sylvaticus? Did this species have more active reproduction in the winter months than in summer? Or is it due to higher mobility in search of food in spring? Can the indifference to moonlight found in this species be associated with a shortage of food resources at a high population density?

Authors: Interpretations of seasonal fluctuations of the species studied are not under the scope of this article, since they were recently reported and analysed elsewhere (ex. Oro et al. 2021, Torre et al. 2022). However, we want to clarify that A.sylvaticus showed a very clear seasonal abundance pattern, because its breeding period extends from autumn to spring in the Mediterranean area, with a summer latency (except in very wet summer periods). This species did not show indifference to moonlight, in fact, was the only species with a negative response to moon light levels, but the effect of that response was almost negligible since not affecting activity. We think that this species, being the main prey of several predators in the Mediterranean forests, can be responding to moon light levels, but its response can be reduced when moon light can be interfered-lowered by vegetation structure, as was the case in our study area.

Some of these questions are not directly related to the assessment of the influence of moonlight on the activity of small mammals, but due to the lack of explanations, there is a feeling of loose ends and weakness of the evidence base. Of course, we admit that the authors themselves repeatedly point out in the text of the manuscript that the parameters under consideration are, to some degree, relative.

Nevertheless, I believe that the manuscript should be published after making small additions.

Kind regards

Reviewer 4 Report

Dear Authors

I have read your paper with a huge interest. My only remarks do not concern the methodology, that I assessed highly, but rather some technical issues. Remember to close brackets - please check lines 42&43, 46&47, 55&56, 90&91, 99.

Figure 2 - the figure should be prepare one more time in better resolution. Legend and the text: natural parks in small letters.

Close to scale bar - instead of "Kilometres" enough will be "km"

Line 143 - flights - to my knowledge it is named differently, maybe like exposition, please check it.

Methodology is described with details.

Compare order of small mammals on the left in figure 3 and 4. I would do the same order in both figures. Or maybe even prepare it on the pattern of tables?

You should prepare a chapter called "Conclusions" with some future directions or remarks or ideas how it is possible to use obtained results.

Author Response

Dear Authors

I have read your paper with a huge interest. My only remarks do not concern the methodology, that I assessed highly, but rather some technical issues. Remember to close brackets - please check lines 42&43, 46&47, 55&56, 90&91, 99.

Authors: Thanks for paying attention to details!!!

Figure 2 - the figure should be prepare one more time in better resolution. Legend and the text: natural parks in small letters.

Authors: We modified the figure by increasing the size of the text

Close to scale bar - instead of "Kilometres" enough will be "km"

Authors: Done

Line 143 - flights - to my knowledge it is named differently, maybe like exposition, please check it.

Authors: Flight is the correct word because Lidar were measured from aeroplanes.

Methodology is described with details.

Compare order of small mammals on the left in figure 3 and 4. I would do the same order in both figures. Or maybe even prepare it on the pattern of tables?

Authors: Thanks, this was a mistake, the order should be from more to less abundant, A.sylvaticus, C.russula, and M.spretus. We changed it in figure 3, following the same order as in Fig.4 and both tables.

You should prepare a chapter called "Conclusions" with some future directions or remarks or ideas how it is possible to use obtained results.

Authors: Done, thanks, rev. 2 also suggested this.

Round 2

Reviewer 1 Report

Dear Authors,

I am satisfied for how you dealt with my comments/suggestions. I think that the MS is now sufficiently improved as for the description of the methods and the possible shortcomings following the approach of retrospectively analysing the effects of moonlight on animals.

Author Response

Thankns for your positive comments!!!

Reviewer 2 Report

Authors amended the Manuscript following all of my previous comments. Now, the paper is almost ok to be published, but I would like to ask authors to consider also another manuscript, showing that, when the sense of safety increases, small rodents may change their behaviour in relation to moon phases:

Viviano A., Mazza G., Di Lorenzo T.., Mori E. (2022). Housed in a lodge: occurrence of animal species within Eurasian beaver constructions in Central Italy. European Journal of Wildlife Research, 68: 75.

Author Response

Thanks, we included the reference suggested.